# Saddle Pressures Factors in Road and Off-Road Cyclists of Both Genders: A Narrative Review

**DOI:** 10.3390/jfmk8020071

**Published:** 2023-05-25

**Authors:** Domenico Savio Salvatore Vicari, Antonino Patti, Valerio Giustino, Flavia Figlioli, Giuseppe Alamia, Antonio Palma, Antonino Bianco

**Affiliations:** 1Sport and Exercise Sciences Research Unit, Department of Psychology, Educational Science and Human Movement, University of Palermo, 90141 Palermo, Italy; 2Regional Sports School of CONI Sicilia, 90141 Palermo, Italy

**Keywords:** biomechanics, bicycle, cyclists, saddle pressure, perineal pressure, urogenital system, injury prevention

## Abstract

The contact point of the pelvis with the saddle of the bicycle could generate abnormal pressure, which could lead to injuries to the perineum in cyclists. The aim of this review was to summarize in a narrative way the current literature on the saddle pressures and to present the factors that influence saddle pressures in order to prevent injury risk in road and off-road cyclists of both genders. We searched the PubMed database to identify English-language sources, using the following terms: “saddle pressures”, “pressure mapping”, “saddle design” AND “cycling”. We also searched the bibliographies of the retrieved articles. Saddle pressures are influenced by factors such as sitting time on the bike, pedaling intensity, pedaling frequency, trunk and hand position, handlebars position, saddle design, saddle height, padded shorts, and gender. The jolts of the perineum on the saddle, especially on mountain bikes, generate intermittent pressures, which represent a risk factor for various pathologies of the urogenital system. This review highlights the importance of considering these factors that influence saddle pressures in order to prevent urogenital system injuries in cyclists.

## 1. Introduction

Cycling is one of the most popular sports in the world and plays an important role in promoting public health, even for those who practice it as an amateur [1,2]. Indeed, cycling, in all its forms and in all its disciplines, is a sport that involves many participants, both at an amateur and professional level.

Cycling is practiced not only on the road, but also off-road and on the track. Four cycling disciplines are part of the Olympics: road cycling, mountain biking, track cycling, and BMX. Road cycling is probably the most widespread and followed discipline of cycling. The races can be one day or in stages. Cross country is the only Olympic discipline of mountain biking. The races take place on undulating mixed circuits with technical descents, forest roads, rock gardens, and obstacles (even artificial) from 4 to 6 km, and the route is repeated several times, depending on the travel time per lap, for a total time race of ≈120 min [3]. Track cycling features several races held within a velodrome, consisting of two straights and two steep curves. BMX is a cycling discipline that uses a small and resistant bike suitable for performing various stunts.

Very often, young people approach the bicycle with a mountain bike, attracted by the sense of freedom and escape from the daily chaos that derives from its use. In fact, the mountain bike, having wider and knobby tires and suspensions, offers the possibility of pedaling on uneven dirt paths and in close contact with nature.

It is important to note that a suitable position on the bike guarantees a good joint dynamic balance, correct breathing, excellent aerodynamics, and adequate weight distribution, favoring maximum muscle work without compromising the optimal balance of the physiology of the entire musculoskeletal system [4,5]. Bicycle fitting is the adjustment of the bicycle to the physical and performance requirements of the cyclist in order to meet the cyclist’s goals and needs. An adequate bicycle fitting, performed by the bicycle fitter, can positively influence performance and the perception of comfort and reduce the risk of injury [4,6].

Indeed, regular cycling has an effect on the prevention of many chronic diseases (e.g., cardiovascular disease, diabetes, cancer, hypertension, obesity, depression, and osteoporosis) [2,7,8]. However, cycling can lead to injuries, most of which affect the urogenital system, causing genital numbness, erectile dysfunction (ED), priapism, infertility, and hematuria and affecting serum prostate-specific antigen (PSA) levels [9]. 

The man–bicycle combination is achieved through five points of contact, such as the two hands on the handlebars, the two feet on the pedals, and the pelvis on the saddle. Pressures are generated at these points of contact given by the distribution of the cyclist’s weight. Saddle pressures are generated when the cyclist is seated, and, of course, no perineal pressure is generated when pedaling in standing position. In fact, the study by Sommer et al. (2001) [10] confirmed that pedaling in an upright position does not cause any alteration of the penile blood supply. As a matter of fact, the constant pressure exerted on the bicycle saddle could be the cause of non-traumatic injuries [11], ranging from saddle sores to more serious disorders related to the urogenital system [12]. 

Perineal discomfort or pain is very common on a bicycle and can be caused by various factors that can affect the normal distribution of pressure in the saddle. As a matter of fact, the saddle is recognized as the major extrinsic risk factor in the development of seat discomfort and perineal pathologies during cycling [13].

Analyzing the pressures in the saddle and controlling the factors that influence it can limit the risk of injury. In the last decade, the measurement of pressures in the saddle and the related compression on the perineum has been investigated [14]. A survey of 2,774 cyclists and 1,158 non-cyclists revealed that cycling is associated with a significantly higher risk of perineal numbness and urethral stricture development [12]. Up to 91% of cyclists experienced perineal numbness. The incidence of erectile dysfunction, especially in long-distance cyclists, was found to be 13%, which is significantly higher than in the general population [15]. Overall, 19% of cyclists who had a weekly distance of more than 400 km complained of erectile dysfunction. Moreover, perineal numbness was reported by 61% of cyclists.

In several sports, the analysis of the pressures in the different points of contact between the athlete and a surface is now widespread. For example, in running athletes, given the increasing attention to the relationship between shoes and running biomechanics [16,17], insole-based sensors that record pressures represent a methodology used to study foot–shoe interactions [18]. Similarly, other studies have investigated the pressure exerted on the saddle on horses and the correlation with back pain in riding horses [19]. Similarly, other studies investigated the differences in plantar pressure distribution between athletes of different sports [20] and between different technical gestures in athletes who practice the same sport [21], and the effects of training or technique [22].

Hence, the need to study saddle pressures in cyclists is of fundamental importance. This review aimed to summarize in a narrative way the current literature on the saddle pressures and the factors that influence saddle pressures in road and off-road cyclists of both genders.

## 2. Methods 

For this review, we searched the PubMed database to identify English-language sources using the following terms: “saddle pressures”, “pressure mapping”, “saddle design” AND “cycling”. We also searched the bibliographies of the retrieved articles. We have included only English-language articles, with no publication time limit.

## 3. Saddle Pressures Measurement

With technological progress, new instruments have been developed to optimize bicycle fitting. The measurement of pressures distribution in the saddle is essential because it analyzes the points of contact of the pelvis, where there are very delicate biological tissues, on the saddle. Saddle pressures are measured using an instrument applied directly to the saddle and equipped with sensors. The instrument allows the evaluation of the contact area, the distribution of the contact pressures, the distance between the ischial tuberosities, and the intermittence of the contact pressures. In detail, the contact area detects the surface of the saddle covered by the pelvis. The distribution of the contact pressures detects the region with a higher or lower load on the saddle. Among these parameters belong the vertical force (i.e., the integral of the pressure applied to the saddle area frame by frame and normalized to the cyclist’s body weight), the mean pressure (i.e., the mean of the pressure values across all mask sensors), and the peak pressure (i.e., the highest value among the pressures recorded) [23]. Once the ischial tuberosities have been detected, the distance between the ischial tuberosities, i.e., how far apart they are from each other, is measured. The intermittence of the contact pressures represents the intermittent pressures exerted by the jerks of the pelvis on the saddle. In fact, the evaluation of the pressures in the saddle can take place in a static or dynamic mode. In static, saddle pressures are recorded with cyclists sitting on the saddle, but without pedaling [24]. In dynamic, the measurement of saddle pressures takes place while the cyclist pedals at a defined pedaling frequency and intensity.

## 4. High Saddle Pressures and Risk of Injury 

In recent years, the scientific literature has focused on the analysis of the mechanical causes of overload injuries affecting the urogenital system in cyclists [23]. The incidence of bicycle-related urogenital symptoms varies [9]. The onset of some disorders is only occasional [25,26], while others, in particular those associated with perineal compression, are more frequent [9]. Sommer et al. (2001) reported an incidence of genital numbness of 61% and erectile dysfunction of 24% in male cyclists whose weekly training exceeded 400 km [10]. Few studies have analyzed the incidence of genital and pelvic floor symptoms in female cyclists [27]. Nonetheless, there is evidence that female cyclists suffer from similar problems as male cyclists, ranging from minor skin lesions to serious conditions, such as pain and neuropathies [27]. The cause of both genital numbness and erectile dysfunction appears to be compression of the pudendal nerve during pedaling [23]. Considering that excessive pressures leads to transient hypoxemia of the nerve [28], the duration of these compressions seems to be more relevant than the intensity of the pressure itself [9,29]. Several studies in the literature show that excessive pressures in the anterior region of the saddle are harmful to erectile tissues compared to pressures recorded in the posterior region of the saddle [30,31]. For this reason, reducing the compressive load on soft tissues is the main goal for the development of bicycle saddle geometries [32].

The factors that influence the distribution of pressures in the saddle are analyzed in the following paragraphs and are shown in Table 1.

## 5. Influence of Sitting Time on the Bike on Saddle Pressures

Cyclists, while pedaling, are not always sitting in the saddle. In fact, very often cyclists pedal standing up, and this factor is given by personal attitudes, the characteristics of the route, and the specialty practiced.

In fact, mountain bikers frequently find themselves in an upright position for very steep and bumpy climbs, where it is necessary to release greater power peaks to tackle them, and for very long descents, where the cyclist is not sitting in the saddle to overcome obstacles.

Conversely, road cyclists spend more time sitting in the saddle due to the regularity of their routes and the lack of obstacles to overcome. Professional road cyclists cover approximately 30,000 to 35,000 km per year between training and competition, and some races, such as the Tour de France, last 21 days (~100 h of racing), during which cyclists cover more than 3500 km [40].

Time spent sitting in the saddle could be a factor influencing saddle pressures and related urogenital pathologies. A few studies in the literature have analyzed saddle pressures for a long time in a sitting position. Often, pedaling standing up and taking breaks during the activity are good habits to reduce pressure on the saddle and prevent the onset of urogenital pathologies [39]. In this sense, off-road cyclists have an advantage over road cyclists because the discipline itself often invites them to pedal standing up.

It could be interesting to evaluate the pressure in the saddle over time to know the maximum time window in which it is possible to remain seated in the saddle while pedaling while maintaining comfort.

## 6. Influence of Pedaling Intensity on Saddle Pressures 

Several studies measuring saddle pressures in a dynamic mode confirmed that pedaling intensity could be an influencing factor.

Carpes et al. (2009) [33] evaluated the effects of 2 different pedaling intensities (150 W and 300 W) and 2 saddle designs (plain saddle and holed saddle) on saddle pressure in male and female cyclists, showing that with the plain saddle, the mean pressure increased with increasing pedaling intensity in males. Furthermore, using the holed saddle, the mean pressure increased with increasing pedaling intensity, regardless of gender.

The study by Bressel et al. (2005) [34] investigated pressures at different pedaling intensities (118 W and 300 ± 82.4 W) and different hand positions (tops and drops) in male and female cyclists. The authors found decreased pressures over most regions of the saddle at higher pedaling intensities and the drop hand position. This may have occurred due to a partial weight shift from the saddle to the pedals as pedaling intensity is increased.

## 7. Influence of Pedaling Frequency on Saddle Pressures

Pedaling frequency could vary the pressure distribution in the saddle. A very high pedaling frequency could lead to higher pressure values in the saddle. In this case, it is advisable to change to a higher gear [39].

## 8. Influence of Trunk Position and Hand Position on Saddle Pressures

The position of the cyclist’s trunk and hand could influence the pressure distribution in the saddle. Carpes et al. (2009) [13] analyzed the effects of trunk position on the pressures exerted in two saddles with different designs in recreational cyclists of both genders seated in a bicycle in static position. In the latter study, saddle pressures were measured on two saddle models (i.e., with and without holes) and in two trunk positions (i.e., upright and forward). The results of this study showed no significant differences in saddle pressures between the two trunk positions in females. Instead, a significant difference was found between trunk positions using the saddle with holes in males. In detail, the forward position of the trunk led to lower pressure using the saddle with holes.

In contrast, the study by Potter et al. (2008) [31] that investigated the influence of gender and hand position on saddle pressures showed that when cyclists pedal and move hand position from the tops to drops, the centers of pressure in all regions move forward, the normalized force and maximum pressure on the posterior region decrease, and female cyclists show an increase in normalized force and maximum pressure in the anterior region. 

Bressel et al. (2003) [38] showed an undesirable pressure to the anterior perineum when the cyclist leans forward on the handlebar of the bicycle.

In another study by Bressel et al. (2005) [34], the authors found that the drop hand position decreased saddle pressures in most regions. This could have occurred due to a partial shift of the cyclists’ weight from the saddle to the handlebars as a result of moving from the top to drop the position of the hands on the handlebars.

However, it has been suggested that body weight bearing on the saddle at the level of the ischial tuberosities is an effective strategy to minimize pressure on the perineum [32]. 

## 9. Influence of Handlebars Position on Saddle Pressures

In the study by Partin et al. (2012) [37], it was demonstrated that the position of the handlebars can influence the pressures in the saddle of female cyclists. Three different handlebar-to-saddle heights were evaluated (at saddle level, lower than saddle, or higher than saddle), and low handlebar-to-saddle placement was associated with greater perineal saddle pressure.

## 10. Influence of Saddle Design on Saddle Pressures

Several studies have investigated the effects of saddle type on saddle pressures. Indeed, the saddle appears to be the main risk factor for urogenital problems. This aspect has contributed to the development of saddles with new designs.

Potter et al. (2008) [31] evaluated the influence of two saddle designs (saddle A, i.e., gender-neutral; saddle B, i.e., female-specific) by adopting two different hand positions in female cyclists. Using saddle B showed a reduction in the normalized maximum anterior pressure with drop hand position compared to saddle A. No significant differences were found between saddles in the maximum pressure with tops hand position or in the posterior region with drop hand position.

The study by Carpes et al. (2009) [33] investigated the influence on pressure in 2 saddles with different designs (plain saddle and holed saddle) at 2 pedaling intensities (150 W and 300 W) in male and female cyclists. The results of this study showed that the saddle design had no effect on pressures, with the only significant difference found when the peak pressure was compared between the holed saddle and the plain saddle during a pedaling intensity of 150 W in females. Higher pressures can be caused by contact with small surfaces on the edge of the saddle hole. On the plain saddle, the pressures were distributed over the entire surface, while on the holed saddle, there were points of higher pressure, which were given by the ischial tuberosities. This last aspect indicates that the contact of the body weight was mainly on the ischial tuberosities, reducing the pressures on the perineal area.

## 11. Influence of Saddle Height on Saddle Pressures

The optimal saddle height is an important aspect for a functional bike fitting [4]. The effects of changes in saddle height on the influence of perineal pressures and discomfort and pain have not been extensively studied. The study by De Looze et al. (2003) [41] demonstrated that pressure distribution and muscle activation are parameters related to discomfort. Setting an appropriate saddle height is very important to increase comfort, prevent injuries, make pedaling more efficient, and improve performance [36,42]. The scientific literature reports different methods to adjust the height of the saddle [42]. Some of these take into account the anthropometric measurements of the cyclist, and others use 2D or 3D kinematic analysis evaluating the range of the joint angle [43]. Christiaans et al. (1998) [5] suggested that the best saddle height for comfort was 106% of the inseam height in men and 107% of the inseam height in women. Another study by Scoz et al. (2022) [43] evaluated subjects with a standardized bicycle fitting procedure based on 3D kinematic data. The results of this study demonstrated that fitting across 15 angular ranges (minimum ankle, maximal ankle, ankle range, ankle angle at bottom, maximum knee flexion, maximum knee extension, knee angle range, knee forward of foot, knee forward of spindle, knee travel tilt, knee lateral tilt, knee lateral travel, hip angle closed, hip angle open, hip angle range, hip lateral travel, back angle, shoulder angle to wrist, shoulder angle to elbow) was sufficient to produce large, long-term effects on pain, fatigue, and comfort. The study by Gamez et al. (2008) investigated the influence of 3 different saddle heights (i.e., 684.1 mm, 709.1 mm, 734.1 mm) on comfort in cyclists. The authors found that the 709.1 mm high saddle was best because the peak pressure distribution and lower-limb muscle activation decreased [35]. The study by Verma et al. (2016) [36] investigated the influence of different saddle positions (neutral, upward, downward, forward, backward) on saddle pressures distribution. The authors found that the discomfort increased with the upward and backward positions compared to neutral. The minimum force became less negative with the forward position compared to the neutral position. The degree of variability of CoP increased in the upward and backward positions.

A recent work by Wang et al. (2020) showed that low saddle height is related to increased knee adduction moments with longer duration in recreational cyclists [44].

Considering that a higher-than-normal saddle height could generate higher pressures, it is very important to adjust the saddle height and the backwardness of the saddle to prevent urogenital pathologies and ensure greater comfort for cyclists.

## 12. Influence of Padded Shorts on Saddle Pressures

A recent study evaluated the influence of various cycling padded shorts on saddle pressures in order to preserve the perineal region and increase the level of comfort [23]. 

A total of 3 padded shorts with different densities were evaluated: basic (60 kg/m^3^), intermediate (80 kg/m^3^), and resistance (80 and 120 kg/m^3^). Saddle pressures were recorded during three trials of twenty minutes each, in which cyclists wore one of the three padded shorts for each trial. The authors found that the vertical force and the mean pressure on the saddle significantly decreased using the basic and endurance padded shorts. Moreover, the peak pressure on the corresponding perineal area of cyclists was significantly increased only with the use of the basic padded shorts. Further, the increase in the length of the center of pressure of the 3 padded shorts at the end of the test suggested that the basic material was starting to lose its elastic properties due to its low foam density (60 kg/m^3^). This study, in line with others in the literature, suggests that it is important to consider cyclists’ padded shorts as a factor influencing saddle pressures [23,45].

## 13. Muscle Activity and Saddle Pressures 

Several studies in the literature have evaluated muscle activity to improve pedaling economy and efficiency [46,47,48,49,50]. The major muscles typically studied were the gluteus maximus (GM), rectus femoris (RF), vastus lateralis (VL), vastus medialis (VM), biceps femoris (BF, long head), medial gastrocnemius (MG), and tibialis anterior (TA) [48]. The study by McDonald et al. (2021) [51] evaluated the muscle activity of these muscles and the changes in the saddle pressure mapping indexes with alteration of the effective seat tube angle (ESTA). To vary the latter, the handlebars and the saddle were moved forwards or backwards simultaneously by 3 cm. The main finding concerns the muscle activity of the BF, GM, and MG, which increased progressively from the forwards to the backwards position. In contrast, the muscle activity of TA decreased progressively from the forwards to the backwards position. No changes were found in the muscle activity of VM, VL, and RF across the different positions. Regarding saddle pressures, a decrease in the percentage of frontal versus rear pressure was found when participants moved from the backwards to the forwards saddle position. A decrease in the mean pubic pressure was also found from the backwards to the forwards position. The results of this study suggest that cyclists in the back position had greater pelvic rotation, therefore increasing perineal pressure and anterior/posterior pressure distribution to compensate. The results were opposite when the saddle was moved forward.

## 14. Gender Differences on Saddle Pressures

The scientific literature shows several studies that have investigated the difference between male and female cyclists in different aspects. The difference in pelvic geometry, power output, and bike fit can influence saddle pressures. For example, the greater width between the ischial tuberosities in female cyclists could reduce the load on the posterior bone structures and increase the load on the perineal region [31]. 

The study by Potter et al. (2008) [31] analyzed gender differences in the distribution of pressures in the saddle. The main aim was to investigate the influence of gender, power, and hand position on saddle pressure distribution. The results confirmed a gender difference in saddle pressures. In detail, with the hand in drop position, only female cyclists showed an increase in the normalized force and maximum pressure in the anterior region.

Similarly, Carpes et al. (2009) also found significant differences between genders with the same workload.

Bressel et al. (2005) [34] showed that females, compared with males, did not show lower peak total, posterior, left, and right pressures in the drop position.

An interesting paper by Hermans et al. (2016) [52] explored the prevalence and duration of urogenital overuse injuries and sexual dysfunctions in female cyclists, showing that after at least 2 h of cycling, dysuria (8.8%), stranguria (22.2%), genital numbness (34.9%), and vulvar discomfort (40.0%) were found.

Coutant-Foulc et al. (2014) [53] reported that female cyclists cycling long distances each week resulted in unilateral swelling of the labium majus.

These findings suggest that it is essential to take into account the gender of cyclists in changes in saddle pressures to prevent perineal injuries.

## 15. Road vs. Off-Road Cyclists: Difference in Saddle Pressures

Between the road bike and the mountain bike, there are substantial differences, both structural and related to the sport, that involve a difference in the variation of saddle pressures. The mountain bike does not allow different grips, such as with the road bike. For this reason, saddle pressures in mountain bike cyclists cannot be influenced by different grips. However, the off-road rough trails cause percussions in the cyclist’s perineum. Genital numbness and erectile dysfunction can result from repeated perineal impacts on the bicycle saddle (micro-trauma).

Sanford et al. (2018) [24] evaluated the relationship between oscillation forces and perineal pressures among cyclists in a simulated laboratory setting by demonstrating a strong linear relationship between oscillation magnitude and perineal pressure during pedaling, mitigated by a saddle post shock absorber.

Studies in the scientific literature report that mountain bikers have a high frequency of extra-testicular and testicular disorders, associated with clinical symptoms in half of the bikers. These may be associated with a high rate of repeated micro-traumas of the scrotal contents [54]. Indeed, mountain bikers are at a higher risk of scrotal disorders than road cyclists [55]. Indeed, a prospective cohort study by Dettori et al. (2004) [56], aimed at exploring the occurrence of erectile dysfunction when using a mountain bicycle/road bicycle after a long-distance cycling event, showed that riding a mountain bicycle increases the risk of erectile dysfunction compared to a road bicycle on a long-distance ride.

Mitterberger et al. (2008) [55] investigated the risk of scrotal disorders in mountain bikers and on-road cyclists. The results showed that 94% of mountain bikers and 48% of on-road cyclists presented with abnormal findings on a scrotal ultrasound.

To the best of our knowledge, there are very few studies that have analyzed saddle pressures during outdoor mountain biking. In mountain biking, there are many factors that can influence the pressure in the saddle. Factors such as tire pressure, shock pressure, shock type, number of shock absorbers, terrain, climbing, and saddle contact time must be considered. Some of these factors are difficult to evaluate in the laboratory on cycle simulators, which is why it is interesting to evaluate the evolution of pressure in the saddle during outdoor pedaling on a mountain bike.

Few studies have so far analyzed the correlation between pressures in the different contact points, and knowing the pressures on the handlebars, in the saddle, and in the soles of the shoes could give us the possibility to identify postural anomalies of the cyclist and inadequate or inefficient pedaling techniques and to develop protocols to make man–bicycle interactions efficient and comfortable.

## 16. Conclusions

The analysis of saddle pressures is of fundamental importance to prevent the onset of urogenital pathologies in cyclists and to guarantee them a greater perception of comfort. The pressures under the perineal area of cyclists are harmful, and the main objective of bike fitting is to limit them as much as possible. This review suggests the importance of knowing the different factors that can influence the distribution of pressures in the saddle. Among these, technicians should be taking into account the influence of (a) pedaling intensity and frequency, (b) trunk position and hand position, (c) handlebars position, (d) saddle design and saddle height during a bike fit for each cyclist. These factors are individual and depend, among others, on gender, anthropometric measurements, and the type of bike. Moreover, they should recommend the most suitable type of padded shorts to use and give advice, such as getting up from the saddle while pedaling in the case of long periods of sitting time.

Considering which factors influence pressure in the saddle is useful for cycling practitioners. In particular, the use of devices capable of recording pressure in the saddle by technicians could make pedaling more efficient and comfortable. In fact, limiting pressure in the pubic region is the main objective to ensure adequate comfort for cyclists.

All the variables that can influence pressure in the saddle analyzed in this review represent risk factors for urogenital pathologies in cyclists. When fitting a cyclist’s bike, these factors should be analyzed, taking into consideration other parameters, such as the weight, age, and level of the cyclist.

Professional bike fitters, cyclists, and trainers can use these findings to improve pedaling [43]. 

## Figures and Tables

**Table 1 jfmk-08-00071-t001:** Factors influencing saddle pressure distributions.

First Author	Year	Measurements	Main Results
Influence of pedaling on saddle pressures			
Carpes, F.P. [33]	2009	Influence of 2 different pedaling intensities (150 W and 300 W) on saddle pressures in male and female cyclists.	The mean pressure increased with increasing pedaling intensity in males. Furthermore, using the holed saddle, the mean pressure increased with increasing pedaling intensity, regardless of gender.
Bressel, E. [34]	2005	Influence of different pedaling intensities (118 W and 300 ± 82.4 W) with different hand position (tops and drops) on saddle pressures in male and female cyclists.	The pressures decreased over most regions of the saddle at higher pedaling intensities and drop hand position.
Influence of saddle on saddle pressures			
Potter, J.J. [31]	2008	Influence of two saddle designs (gender-neutral and female-specific) with two different hand positions in female cyclists.	Using the female-specific saddle, the normalized maximum anterior pressure with drop hand position decreased compared to the gender-neutral saddle. No significant differences were found between saddles in the maximum pressure with tops hand position or in the posterior region with drop hand position.
Carpes, F.P. [33]	2009	Influence of 2 saddle designs (plain saddle and holed saddle) at 2 pedaling intensities (150 W and 300 W) in male and female cyclists.	The saddle design had no effect on pressures, with the only significant difference found when the peak pressure was compared between the holed saddle and the plain saddle during a pedaling intensity of 150 W in females.
Gámez, J. [35]	2008	Influence of three different saddle heights on comfort in cyclists (measuring saddle pressures).	The best level of comfort was the intermediate one; with this height, the peak pressure distribution and activation of lower limb muscles decreased.
Verma, R. [36]	2016	Influence of five different saddle positions (neutral, upward, downward, forward, and backward) on comfort in cyclists (measuring saddle pressures).	The discomfort increased with upward and backward positions compared to neutral. The minimum force became less negative with forward position compared to neutral position. The degree of variability of CoP increased in the upward and backward position.
Influence of handlebars on saddle pressures			
Partin, S.N. [37]	2012	Influence of three different handlebar-to-saddle heights (at saddle level, lower than saddle, or higher than saddle) on saddle pressures.	Low handlebar-to-saddle placement was associated with greater perineal saddle pressure.
Influence of trunk position and hand position on saddle pressures			
Carpes, F.P. [13]	2009	Effects of two trunk positions (upright and forward) on the pressures exerted in two saddles with different designs.	No significant differences in saddles pressures between the two trunk positions in females. The forward position of the trunk led to lower pressures using the saddle with holes in males.
Potter, J.J. [31]	2008	Influence of two hand positions (tops and drops) on saddle pressures.	From tops to drops hand position, the CoP moved forward, the normalized force and maximum pressure on the posterior region decreased. Only female cyclists showed an increased normalized force and maximum pressure in the anterior region.
Bressel, E. [38]	2003	Influence of hand position (tops and drops) on saddle pressures.	With the cyclist forward on the handlebars (drops position), there was an undesirable anterior pressure at the perineum.
Bressel, E. [34]	2005	Influence of hand position (tops and drops) on saddle pressures.	The drops hand position decreased saddle pressures in most regions.
Influence of padded shorts on saddle pressures			
Marcolin, G. [23]	2015	Influence of three padded shorts with different densities (basic, intermediate, resistance) on saddle pressures.	The vertical force and mean pressure on the saddle significantly decreased using the basic and endurance padded shorts. The peak pressure on the perineal area significantly increased only using the basic padded shorts.
Influence of sitting time on the bike on saddle pressures			
Bond, R.E. [39]	1975	Influence of sitting time or standing up on pedals on saddle pressures.	Often, pedaling standing up and taking breaks during pedaling are good habits to reduce saddle pressures and prevent the onset of urogenital pathologies.

Legend: W, Watt; CoP, Centre of Pressure.

## Data Availability

Not applicable.

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
