# Peer review of "Saddle Pressures Factors in Road and Off-Road Cyclists of Both Genders: A Narrative Review"

_jfmk, 2023, doi:10.3390/jfmk8020071_

Round 1

Reviewer 1 Report

This is a very interesting review work. It is well written and explicit.

I think the manuscript can be accepted with only minor revisions.

When citing other authors res by writing XXXet al., et al. should be in italics.

Used al the units in International System (SI), line 105.

Author Response

Reviewer #1

Comments and Suggestions for Authors

This is a very interesting review work. It is well written and explicit.

I think the manuscript can be accepted with only minor revisions.

Author response: Dear Reviewer, thank you for appreciating our study.

When citing other authors res by writing XXXet al., et al. should be in italics.

Author response: Dear Reviewer, thank you for paying attention to our forgetfulness. We have provided to report "et al" in italics throughout the manuscript.

Used al the units in International System (SI), line 105.

Author response: We agree with the reviewer. We are sorry for the mistake.

Reviewer 2 Report

This study discusses about saddle pressure factors in road and off-road cyclists of both genders based on literature review. The result of this study is of interest for researchers in this field. The manuscript is also well written. Following suggestion may improve the impact of the paper.

l  Is there any relationship between muscle activation and saddle pressure? A review about this matter is welcome.

Author Response

Reviewer #2

Comments and Suggestions for Authors

This study discusses about saddle pressure factors in road and off-road cyclists of both genders based on literature review. The result of this study is of interest for researchers in this field. The manuscript is also well written. Following suggestion may improve the impact of the paper.

Author response: Dear Reviewer, thank you for the evaluation of our paper.

Is there any relationship between muscle activation and saddle pressure? A review about this matter is welcome.

Author response: Dear Reviewer, thank you for paying attention to this interesting aspect. We reviewed the literature on this topic and added a paragraph in the manuscript entitled “Muscle activity and saddle pressures”. Thank you for the suggestion.

Reviewer 3 Report

The topic is interesting and undoubtedly important, worthy of publication. I appreciate the work and the scope of knowledge. However, it is necessary to reconstruct the article in order to establish a logical sequence of works. The structure of the work is not entirely consistent with the review article. This causes some confusion and misleads the reader; it is not clear what the results of this work are.

The article needs to be improved in its structure and separation of individual chapters.

The conclusions could be more specific about the factors discussed.

The structure of a review article typically includes the following sections: Introduction - where the authors introduce the reader to the subject of the review article, explaining why it is important and what will be discussed. Method – in which the authors describe how they conducted the literature review, what criteria they used to select studies for analysis, what databases they searched and what searches they used. Results - where the authors summarize and synthesize the results of their literature review, highlighting the main themes and patterns in the studies that have been included. Discussion – where the authors discuss the significance of their results and what conclusions can be drawn, what research questions remain open, and what implications for practice may arise from their work. Conclusion - where the authors summarize the key points of their paper and indicate what further research may be needed. References - where authors list all publications cited in the review article. In addition, review articles may contain sections such as: Definitions of key terms - if the concepts used in the review article are complex or require additional explanation. Inclusion/exclusion criteria for studies – if the literature review covered only certain types of studies or had specific inclusion/exclusion criteria. Limitations - where the authors discuss the limitations of their literature review and what implications this has for their results and conclusions. Perspectives - where the authors indicate what are the further directions of research in the field covered by the review article

Author Response

Reviewer #3

Comments and Suggestions for Authors

The topic is interesting and undoubtedly important, worthy of publication. I appreciate the work and the scope of knowledge. However, it is necessary to reconstruct the article in order to establish a logical sequence of works. The structure of the work is not entirely consistent with the review article. This causes some confusion and misleads the reader; it is not clear what the results of this work are.

The article needs to be improved in its structure and separation of individual chapters.

The conclusions could be more specific about the factors discussed.

The structure of a review article typically includes the following sections: Introduction - where the authors introduce the reader to the subject of the review article, explaining why it is important and what will be discussed. Method – in which the authors describe how they conducted the literature review, what criteria they used to select studies for analysis, what databases they searched and what searches they used. Results - where the authors summarize and synthesize the results of their literature review, highlighting the main themes and patterns in the studies that have been included. Discussion – where the authors discuss the significance of their results and what conclusions can be drawn, what research questions remain open, and what implications for practice may arise from their work. Conclusion - where the authors summarize the key points of their paper and indicate what further research may be needed. References - where authors list all publications cited in the review article. In addition, review articles may contain sections such as: Definitions of key terms - if the concepts used in the review article are complex or require additional explanation. Inclusion/exclusion criteria for studies – if the literature review covered only certain types of studies or had specific inclusion/exclusion criteria. Limitations - where the authors discuss the limitations of their literature review and what implications this has for their results and conclusions. Perspectives - where the authors indicate what are the further directions of research in the field covered by the review article.

Author response: Dear Reviewer, thank you for your comment. We agree that the structure of an article requires an academic methodology and that, typically, the sections are those described in the reviewer's comment (Introduction, Methods, Results, Discussion, Conclusion). However, this must be rigorously developed in the case of a systematic literature review and is not binding or obligatorily applicable in a narrative review where the aim is to describe in a narrative and non-systematic way a research topic. For example, the "search methodology" is an obligatory section in systematic reviews and meta-analyses but not in narrative literature reviews. Furthermore, this is confirmed by the fact that the journal does not require subdivision into the aforementioned sections for narrative review and this is confirmed by the very numerous narrative reviews already published. Here are some of these:

https://www.mdpi.com/2411-5142/7/3/68

https://www.mdpi.com/2411-5142/7/3/61

https://www.mdpi.com/2411-5142/7/2/33

However, accepting the reviewer's suggestion, considering that in the previous version of the manuscript we had already inserted the "Introduction" and "Conclusions" sections, we have added a brief "Methods" section in which we have reported the databases and the keywords used and the inclusion criteria for articles.

Reviewer 4 Report

Comments to the manuscript with ID: jfmk-2346572 entitled: Saddle Pressures Factors in Road and Off-Road Cyclists of Both Genders: A Narrative Review. This manuscript review the saddle pressures in cyclist and the influence of different variables such as pedaling frequency, trunk position, hand position, biking time, saddle height, gender influence and pad length.

This narrative review is well written and some changes are suggested.

0.- Abstract.

            The abstract explained properly the aim of the narrative review and the highlight are well described.

1.- Keywords

            Too many keywords and similar such as bicycle and bike. Please add less keywords and delete similar

2.- Introduction

            Lines 45 – 48: Please add references

            Lines 50 – 52: Please add references

            Line 71: Is not well written, please review and explain for better comprehension.

            Lines 72 – 73. Please add references.

            Line 101. Please rewrite. Too many saddle in one sentence.

            Lines 123 – 126: These lines are the same as 78 – 80 lines. Rewrite please.

            Lines 171 – 174: Add references

            Why are only 2 tables?? Please add table according to each influence variable on saddle pressure

            Line 234: Setting an appropriate saddle height is only important for comfort? Please clarify.

            Lines 248 – 249: Explain better what are the authors referring to “Intermediate” term.

            Lines 261 – 274: Why is one one reference in Influence padded shorts on saddle pressures?

            How were the inclusion criteria references in the narrative review?

3.- Conclusion

            Should be supported according to the different variables influences. Please rewrite

The grammar english is properly and minor changes are required.

Author Response

Reviewer #4

Comments and Suggestions for Authors

Comments to the manuscript with ID: jfmk-2346572 entitled: Saddle Pressures Factors in Road and Off-Road Cyclists of Both Genders: A Narrative Review. This manuscript review the saddle pressures in cyclist and the influence of different variables such as pedaling frequency, trunk position, hand position, biking time, saddle height, gender influence and pad length.

This narrative review is well written and some changes are suggested.

Author response: Dear Reviewer, thank you for the evaluation of our paper.

0.- Abstract.

            The abstract explained properly the aim of the narrative review and the highlight are well described.

Author response: Dear Reviewer, thank you. We appreciated.

1.- Keywords

            Too many keywords and similar such as bicycle and bike. Please add less keywords and delete similar.

Author response: Dear Reviewer, thank you for your comment. In the new version of the manuscript, we removed the similar keywords as suggested.

2.- Introduction

            Lines 45 – 48: Please add references

            Lines 50 – 52: Please add references

            Line 71: Is not well written, please review and explain for better comprehension.

            Lines 72 – 73. Please add references.

            Line 101. Please rewrite. Too many saddle in one sentence.

            Lines 123 – 126: These lines are the same as 78 – 80 lines. Rewrite please.

            Lines 171 – 174: Add references

Author response: Dear Reviewer, thank you. As suggested, we added all the references and rewrote the sentences indicated.

            Why are only 2 tables?? Please add table according to each influence variable on saddle pressure

Author response: Dear Reviewer, thank you for your insightful comment. In the new version of the manuscript we have included a single table in which we have reported all the factors that influence the pressures in the saddle (that is, all those described in the paragraphs).

            Line 234: Setting an appropriate saddle height is only important for comfort? Please clarify.

Author's response: Dear reviewer, thank you for your suggestion. In the new version of the manuscript, we have clarified the sentence. We reported as follows: “Setting an appropriate saddle height is very important to increase comfort, prevent injuries, make pedaling more efficient, and improve performance”

            Lines 248 – 249: Explain better what are the authors referring to “Intermediate” term.

Author response: Dear reviewer, thank you for your suggestion. We have better expressed the concept of “intermediate” specifying the three different saddle heights (i.e., 684.1 mm, 709.1 mm, 734.1 mm) considered by authors.

            Lines 261 – 274: Why is one one reference in Influence padded shorts on saddle pressures?

Author response: Dear Reviewer, to the best of our knowledge (and considering that we have not performed a systematic literature review) only one study has evaluated the influence of padded shorts on saddle pressures.

            How were the inclusion criteria references in the narrative review?

Author response: Dear Reviewer, thank you for your comment. This, and other sections, must be rigorously reported in the case of a systematic literature review and is not obligatorily applicable in a narrative review where the aim is to describe in a narrative and non-systematic way a research topic. For example, the "search methodology" is an obligatory section in systematic reviews and meta-analyses but not in narrative literature reviews. Furthermore, this is confirmed by the fact that the journal does not require these details for narrative review and this is confirmed by the very numerous narrative reviews already published. Here are some of these:

https://www.mdpi.com/2411-5142/7/3/68

https://www.mdpi.com/2411-5142/7/3/61

https://www.mdpi.com/2411-5142/7/2/33

However, following the reviewer's suggestion, in the new version of the manuscript we have added a brief "Methods" section in which we have reported the databases and the keywords used and the inclusion criteria for articles.

3.- Conclusion

            Should be supported according to the different variables influences. Please rewrite.

Author response: Dear Reviewer, thank you for acute comment. We edited the conclusions by adding specific statements for technicians according to the factors analyzed.